# Attitudes and Associated Demographic Factors Contributing towards the Abuse of Illicit Drugs: A Cross-Sectional Study from Health Care Students in Saudi Arabia

**DOI:** 10.3390/medicina58020322

**Published:** 2022-02-21

**Authors:** Wajid Syed, Ayesha Iqbal, Nasir A. Siddiqui, Ramzi A. Mothana, Omer Noman

**Affiliations:** 1Department of Clinical Pharmacy College of Pharmacy, King Saud University, Riyadh 11451, Saudi Arabia; 2Division of Pharmacy Practice and Policy, School of Pharmacy, University Park Campus, University of Nottingham, Nottingham NG7 2RD, UK; ayesha.iqbal@nottingham.ac.uk; 3Department of Pharmacognosy, College of Pharmacy, King Saud University, Riyadh 11451, Saudi Arabia; nsiddiqui@ksu.edu.sa (N.A.S.); rmothana@ksu.edu.sa (R.A.M.); oalmarfadi@ksu.edu.sa (O.N.)

**Keywords:** illicit drugs, substance abuse, factors, tobacco, hookah

## Abstract

*Background and objective*: The purpose of this study is to compare the attitudes, views, and factors that influence drug abuse among pharmacy and nursing students at a Saudi Arabian university. *Materials and Methods*: A cross-sectional study, was conducted among pharmacy and nursing students who are currently enrolled in the respective courses at the study site. The data were collected over 4 months from August to November 2019 using structured self-administered paper-based questionnaires. *Results*: Among the participants, pharmacy students accounted for 184 (58.2%) while 132 (41.8%) of the students were from nursing. More than a third of the students 129, (40.8%) smoked cigarettes. The majority of pharmacy (80.4%) and nursing students (67.4%) reported having undertaken a drug misuse course in college. Among the participants, 132 (41.7%) stated that an offer from friends, followed by joy seeking 129 (40.8%), parents’ divorce 126 (39.8%), having access to drugs 125 (39.5%), family issues 110 (34.8%), 66 (20.8%) having a family member who is addicted, and 101 (31.9%) reported curiosity to be the factors regarding the use of abusive drugs. Transient euphoria (75.9%) followed by depression 197 (62.3%) was the most prevalent physical or psychological change that occurred following drug use. The family size and father’s education have significantly affected the attitudes scores of the students (F = 5.188; *p* = 0.0001). *Conclusion*: In this study, joy-seeking, access to drugs, and family issues were found to be the major factors listed as reasons for drug abuse, with some of them being controllable or reversible. Educating about the adverse outcomes of abused drugs is warranted.

## 1. Introduction

The use of abusive or prohibited substances remains a growing problem among young individuals and students, which contributes to socioeconomic and humanistic burden worldwide [1,2]. According to the United Nations world drug report, in 2021 globally around 275 million individuals used prohibited substances, and 13% of abusers suffered from drug use disorders [3]. The global prevalence of prohibited substances in 2019 was 5.5% [3,4]. The global statistics report in 2021 stated, over 11 million populations, are estimated to use injectable drugs, most of them were suffering from hepatitis [3,4]. Additionally, a recent study among adolescents in 2021 reported alcohol, marijuana, and nicotine were the most commonly abused drugs. The prevalence of alcohol was 46.5%, while marijuana was 30.5%, nicotine was 26.6% [2]. The most commonly used prohibited substances among teenage students were found to be cannabis (17%) followed by nicotine (17%) [5]. The highest prevalence of using prohibited substances was found in North America including the United States (USA) and Canada [6,7]. Earlier studies report that the use of natural stimulants and other prohibited substances is on the rise in middle eastern countries such as Saudi Arabia [8,9,10].

Several studies have reported that the use of banned substances is motivated by friends’ and/or to reduce the stress associated with parental pressure [11,12,13]. Another survey conducted in the US reported the most common motivations for using illicit drugs in adults were social, recreational purposes, followed by overcoming difficulties and physical requirements [14,15]. Although previous studies showed that amphetamine (4–70.7%), heroin (6.6–83.6%), alcohol (9–70.3%), and cannabis (1–60%) were the most widely used prohibited drugs among Saudi adults [16,17]. Previous reports found that amphetamine was the most widely used prohibited drug among Saudi adults [10,11,18]. Another report stated that in Saudi Arabia, the use of cannabis and amphetamines was on the rise whereas the use of heroin and volatile substances were found to be decreasing [16]. Drugs injections abuse was linked to anxiety, depression, and hepatitis as co-morbid conditions in Saudi Arabia [10,11,16]. Several studies have found that medical and other healthcare students were found to have the highest frequency of using abusive substances [9,12,16,19]. However other studies indicate that the usage of banned drugs and illegal substances varies by age [9], geographical region, and the abusive persons’ previous psychosocial and/or criminal history [20,21]. Other studies have also linked the use of alcohol, cannabis, and other stimulants to a lack of knowledge about prohibited substance use [9,20,21,22]. The higher the education, the lower the abuse to prohibited substances [9,20,21,22,23].

Illicit drugs are substances that are highly addictive and unlawful [16]. Their use was found to cause undesirable and chronic physiological and psychological disorders, which resulted in disability-adjusted life years, and premature mortality [6,16]. In addition, studies also found that chronic use of illicit drugs was associated with an increased risk of multiple types of cancers, and the development of non-malignant respiratory diseases, cardiovascular diseases, as well as a wide array of other chronic health conditions [18]. In Saudi Arabia consumption of abusive, illegal, and prohibited substances is a serious offense, severely punishable according to “Sharia law” [21,24,25], which limits the studies, conducted to examine the drug abuse issue. Nevertheless, attitudes and factors affecting the use of abusive, illegal, and prohibited substances among students and other individuals are an important topic and deserve to be examined. Knowledge and awareness of health in students is linked with the use of abusive drugs [24,25]. Understanding the views will determine the extent of their involvement in the management programs for this serious problem, detrimental to the community health. Therefore, this study aims to evaluate health care student’s attitudes and pre-disposing factors influencing the use of abusive, illegal, and prohibited substances.

## 2. Methods and Materials

### 2.1. Study Design and Settings

The study was designed as a cross-sectional comparative survey, where pharmacy and nursing students who were currently pursuing their graduate programs at King Saud University Riyadh Saudi Arabia from August to November 2020, were invited to participate in this study. Entry-level students, aged ≥18 years who expressed a willingness to complete the survey and from the college of pharmacy and nursing were included in the study. Students below the age of 18 and students from other disciplines were excluded from the study. The data were collected using a self-administered paper-based questionnaire. Ethical approval was obtained from the Research Ethics Committee, College of Medicine, King Saud University Riyadh, Saudi Arabia dated on 21st May 2019 (E-19-3984). Participation in the study was voluntary.

There were approximately 500 senior residential students from both courses at the KSU campus from whom we could obtain the required sample size as calculated with the Raosoft sample size calculator (http://www.raosoft.com/samplesize.html, last accessed on 21 December 2021) with a 95% confidence level and a pre-determined margin of error of ±5%. We assumed that the response distribution for each question would be 50% because we were unsure of the potential results for each question. The calculated sample size was 218, but we decided to survey at least 400 students in an attempt to ensure higher reliability.

### 2.2. Questionnaire Design

The development of the questionnaire for this study was done by modifying a validated survey instrument by Geramian et al. [26]. The questionnaire was initially developed in the English language and the final version was translated into Arabic language. The translation of the questionnaires was done by using forward and backward translation protocols [27]. The translated survey instrument was subjected to face and content validity by a group of experts. Two academics (native Arabic speakers) with extensive experience in preparing and validating survey instruments and one researcher (pharmacist-field expert) from the clinical pharmacy department college of pharmacy validated the data collection instrument. The content was modified in light of the feedback received from the experts. The final version of the survey instrument consisted of 16 questions divided into four sections. The first section (Four items) had questions regarding the demographic characteristics of the participants, including age, family size, father’s educational level, and occupation and the second section (two items) evaluated the students’ smoking status and history of drug abuse education using a binary scale (Yes/No). The third section had one question with multiple-choice options, having a five-point Likert rating scale from 1 to 5 ranging from Not important/Slightly important/fairly important/important/very important. The last section consisted of two questions; the first, was about what physical or psychological changes were experienced after the use of drugs (six items) whereas the second question was about opinions around abusive, illegal, and prohibited substances, three items) on a three-point scale (agree, no comment, disagree).

### 2.3. Sampling and Data Collection

A two-step sampling approach was used for data collection for this study to ensure generalizability and minimize selection bias. First, the college of pharmacy, followed by nursing was selected to collect the data. At the time of the study, the students were in the first, second, and third years of the course. The students of different years, across both disciplines, were randomly invited to participate in the study by a researcher who had experience in conducting research involving human participants. The researcher approached the class leader of each year, in pharmacy and nursing respectively, explained the purpose of this study, and handed over the questionnaires, to be distributed to students. Some of the students completed the questionnaire on the spot, while others asked the time to collect the questionnaire at a mutually agreed time. The completion of the questionnaire was considered as the consent of the participant.

### 2.4. Data Management

Data extraction is the crucial step in the research process and involves careful examination of completely and incompletely answered questionnaires [28]. For the current study, data were checked for accuracy and completeness and any missing responses, incomplete responses, and/or invalid responses were excluded from the study as shown in Figure 1.

### 2.5. Data Analysis

The collected data were analyzed using the IBM SPSS Statistics 22 (IBM Inc., Chicago, IL, USA) and IBM SPSS 26 (IBM Inc., Chicago, IL, USA) software. Descriptive statistics, frequencies, and percentages were used to summarize the data. The mean attitudes scores were calculated for the attitudes questionnaires. The one-way ANOVA and student’s t-test test was used, as appropriate, to assess the association between demographic characteristics. A *p*-value of less than 0.05 was considered statistically significant.

## 3. Results

The response rate for this study was 79%, as 316 out of 400 participants completed and returned the questionnaires. All study participants were enrolled at King Saud University’s nursing and pharmacy colleges, with the vast majority being Saudis. More than half were pharmacy students 184 (58.2%) and the rest 132 (41.8%) were nursing students. Approximately 69% of the pharmacy 130 (70.7%) and 81 (61.4%) of the nursing students who responded to the survey had a family of 5 to 9 members. In total, 147 students’ fathers were employed (46.5%), while 37% of pharmacy students and 23.5% of nursing students’ fathers had retired. In terms of father’s education, more than half of the 105 (57.1%) pharmacy and 78 (59.1%) nursing fathers had received university-level education. About 55% of the pharmacy (*n* = 101) and 65.2% (*n* = 86) of nursing students had never smoked. One hundred forty-eight (80.4%) of pharmacy students and 89 (67.4%) of nursing students reported taking a drug abuse course. Furthermore, demographic details are presented in Table 1.

In regards to participants perception regarding factors contributing toward abusive drug use among students, approximately 132 (41.7%) of total participants agreed that offer from friends is the most important/very important factor for using abusive, illegal, or prohibited substances followed by joy seeking 129 (40.8%), parents’ divorce 126 (39.8%), having access to drugs/substances 125 (39.5%), problems in family 110 (34.8%), presence of an addicted person in the family 66 (20.8%), and teenagers curiosity 101 (31.9%). Further factors and student attitudes toward the use of abusive substances and drugs can be found in Table 2 below.

Regarding identifying what effects could occur immediately following the intake of the abusive or illicit substance and/or drug (within how many hours), the majority of the students, 240 (75.9%), agreed that transient euphoria occurs after the use of illicit drugs. More than half of the students 197 (62.3%) also agreed that depression could be experienced because of illicit substance or drug use. Approximately one-third of participants 145 (45.9%) reported improvement in some somatic diseases (pain, distress), which was followed by increased self-confidence (Table 3). Out of the total, 212 students (67.1%) agreed that it is possible to become addicted (physically or/and psychologically) to drugs or illicit substances even after a single exposure. Furthermore, 58.9% of the 186 students did not agree that occasional use of prohibited drugs was acceptable (Figure 2). However, 146 (46.2%) of students agreed that drugs like hashish are not addictive (Figure-2). The mean attitude score was significantly higher among students of a family of >10 people in comparison to families of 5–9 and 1–4 people (*p* < 0.001). Similarly, the father’s education and occupation were found to be significantly associated with the mean attitude score (*p* < 0.001). The student’s previous experience with a drug abuse course taken during their student life was found to have no impact on the perceived factors resulting in the use of abusive or illicit substances use (t = 0.996; *p* = 0.320) as shown in Table 4. Additionally, the attitude score was not significantly associated with course of study (t = −1.120; *p* = 0.264).

## 4. Discussion

To the best of our knowledge to date, this is the first study that has evaluated the attitudes of health care students toward the factors associated with the use of illicit drugs or prohibited substances among students in Saudi Arabia. Not much literature was identified nationally and internationally about the attitudes of pharmacy and nursing students toward illicit drug use, however, most of the literature reported on universities students’ attitudes toward drug abuse [29,30,31]. (Rahimian Boogar et al. 2014; Al-Shatnawi et al. 2016; Al Ghobain et al.,2016). This study significantly contributes to identifying factors that could contribute to the use of illicit drugs and prohibited substances among the young population and university students in Saudi Arabia and will serve as a baseline study for further studies looking at developing targeted interventions and services to reducing the risk of predisposing factors or helping in the management and withdrawal of addictive substances. Additionally, this investigation considers it significant, since the use of abusive or prohibited substances is indirectly connected to the abused individual’s behavior, which contributes to rational opioid prescribing for dental and pharmacy students, especially highlighting high-risk participants and managing their medications, as well as mental health services by pharmacists, or administering naloxone to addicts in pharmacies.

Despite that most illicit drugs are classified as prescription-only medications, most people take them without a prescription in Saudi Arabia [17]. Therefore, this study provides an understanding of future health care professionals’ attitudes toward abusive drugs and informs their opinions on which factors could be potentially leading to abuse of prescription medications. The study findings show that the majority of the respondents perceived joy seeking as the most important factor contributing to the use of abusive drugs or illicit substances followed by friends’ offers, family disputes, and teenagers’ curiosity to experience them. These results are consistent with the findings of Terry-McEllrath et al. where participants admitted to using illicit substances and abusive drugs to get high, enjoy a good time, and experiment with the drugs [14], additional reasons to abuse drugs or use illicit substances were reported by another study from Saudi Arabia by Al Ghobain et al., where Saudi athletes were found to abuse these substances to enhance their performance to improve social recognition [31]. However, the use of abusive, illicit, or prohibited substances differ in different populations, and Di Luigi L reported people abusing them to obtain therapeutic relief (e.g., medical treatments, surgical procedures) [32].

According to a recent report, genetics, family history, mental health, and environmental stressors can increase substance abuse susceptibility, and the overall risk of using abusive, illicit, or prohibited substances can double [33,34]. However, earlier studies documented that the use of prohibited substances among students was higher particularly smoking marijuana [35,36,37]. Although according to previous evidence, more particular reasons to using illicit drugs were to cope with a negative effect, to be free from worries, to get relax, to be away from problems, to control anger, to increase drug effects, and to cope with physical needs [14]. While the current findings reported that the presence of an addicted person or friend, easy access to drugs, as predisposing factors to abuse drugs and use illicit substances. Similarly, other previous studies reported people with a positive family history of drug use as well as access to controlled drugs were more likely to have substance use-related behavioral problems [30,38,39].

In this study, most of the health care students practice tobacco smoking, which was similar to previous studies published in developed and developing countries [30,40,41,42,43]. However previous studies found the use of illicit drugs such as marijuana, prescription stimulants, opioids, cocaine, ecstasy by pharmacy students [30,40,42,43]. Other drug use, such as sedatives, hallucinogens, anxiolytics, heroin, and cocaine, was also present in students, as documented in prior investigations. Even though today’s students will become tomorrow’s professionals, priority should be given to addressing these harmful behaviors to avoid circumstances that could jeopardize future health care processes and outcomes.

Similarly, previous studies evidenced that attitude scores were significantly different among nicotine and alcohol users [42,43]. The study findings show that students’ or parents’ moral standards, as well as their proclivity for substance abuse, can influence a person’s attitudes toward addicts. This is similar to other studies where gender (male) and parent’s education were one of the most important elements influencing views about drugs and drug addicts [44,45,46,47,48,49]. Another study from Turkey report similar findings, where parents’ education was significantly associated with student’s mean attitude scores [43].

We encourage the implementation of preventive and treatment interventions and public health strategies. Developing and establishing training for students in the health professions about abusive, illicit drugs, and prohibited substances might offer additional skills to these professionals, which could help identify drug-seeking behavior and identify high-risk people in society [44,45].

There are certain limitations to the current study. First, the findings were based on a self-administered questionnaire, which could have increased the risk of biases such as social desirability bias or recall bias. Second, the findings came from a single Saudi Arabian university, making them unrepresentative of others and ungeneralizable globally. Third, due to the better access to male students while disseminating the questionnaire, the study did not include female students. Fourth, due to differences in populations, study design, and techniques used to measure attitude, the parameters mentioned in this study may differ from those mentioned in other studies. Lastly, this study did not involve participants from a normal population of similar ages to compare attitudes and consequences, which might be different. Despite these limitations, our research suggests that more emphasis be placed on increasing students’ awareness of prohibited substances and correcting misconceptions about using them for pleasure or stress relief to make them more competent in raising public awareness about prohibited substances.

## 5. Conclusions

Our study highlights the attitudes and perceptions of students toward factors contributing to drug abuse. According to findings joy-seeking, access to drugs, and family issues were found to be the major factors listed as reasons for drug abuse, with some of them being controllable or reversible. More importantly, the prevalence of abused drugs is on the rise compared to both national and international studies, which may potentially associate with negative consequences on public health. The factors identified in this study present an avenue of opportunities, services, or interventions to be designed and implemented to target reducing and overcoming this issue. Therefore, we advocate the implementation of educational programs that teach students about the harmful outcomes of abusive drugs and avoid their complications. Additionally, the current findings could serve as support for faculties of pharmacy and nursing to improve their curriculum to create awareness and to encourage to prevent the use of prohibited substances, also support the development of messages aimed at safely using drugs.

## Figures and Tables

**Figure 1 medicina-58-00322-f001:**
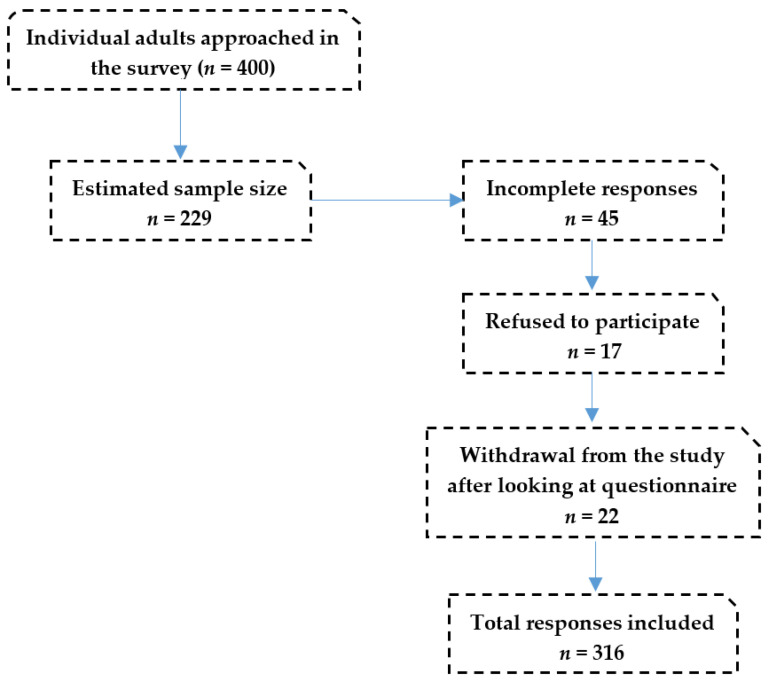
Flowchart of participant responses.

**Figure 2 medicina-58-00322-f002:**
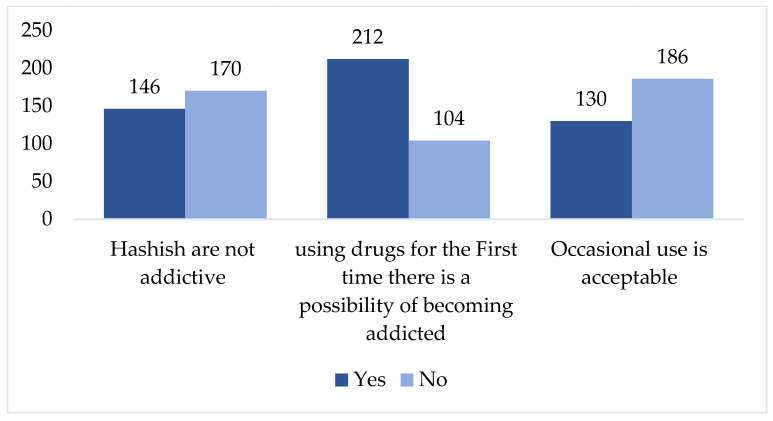
Opinions/views of students about addictive drugs.

**Table 1 medicina-58-00322-t001:** Demographics of the participants (*n* = 316).

Participant Characteristics	Pharmacy *n* (%)	Nursing *n* (%)	Total *n* (%)
Family size			
1–4	35 (19%)	38 (28.8%)	73 (23.1%)
5–9	130 (70.7%)	81 (61.4%)	211 (66.8%)
> 10	19 (10.3%)	13 (9.8%)	32 (10.1%)
Course of study	184 (58.2%)	132 (41.7%)	316 (100)
Father education			
Illiterate	5 (2.7%)	17 (12.9%)	22 (7.0%)
Able to read and write	11 (6%)	5 (3.8%)	16 (5.1%)
School education	63 (34.2%)	32 (24.2%)	95 (30.1%)
University-level education	105 (57.1%)	78 (59.1%)	183 (57.9%)
Father occupation			
Clerk	77 (41.8%)	70 (53%)	147 (46.5%)
Self-employed	15 (8.2%)	23 (17.4%)	38.0 (12%)
Unemployed	8 (4.3%)	4.0 (3%)	12.0 (3.8%)
Retired	68 (37%)	31.0 (23.5%)	99 (31.3%)
Others	16 (8.7%)	4.0 (3.0%)	20.0 (6.3%)
Smoking			
Yes	83 (45.1%)	46 (34.8%)	129 (40.8%)
no	101 (54.9%)	86 (65.2%)	187 (59.2%)
Received any course/education about drug use in the college			
Yes	148 (80.4%)	89 (67.4%)	237 (75%)
no	36 (19.6%)	43 (32.6%)	79 (25%)

**Table 2 medicina-58-00322-t002:** Attitudes of students towards factors contributing to the use of illicit drugs and substances among participants.

Variables	Not Important *n* (%)	Slightly Important *n* (%)	Fairly Important *n* (%)	Important *n* (%)	Very Important *n* (%)	Mean ± SD
Teenagers curiosity	119 (37.7)	28 (8.9)	68 (21.5)	25 (7.9)	76 (24.1)	2.71 ± 1.60
Joy-seeking	89 (28.2)	50 (15.8)	48 (15.2)	55 (17.4)	74 (23.4)	2.92 ± 1.54
Somatic diseases	170 (53.8)	57 (18.0)	37 (11.7)	20 (6.3)	32 (10.1)	2.009 ± 1.35
Mental disorder	112 (35.4)	60 (19)	44 (13.9)	43 (13.6)	57 (18)	2.59 ± 1.52
Lack of knowledge about complications	119 (37.7)	42 (13.3)	59 (18.7)	40 (12.7)	56 (17.7)	2.59 ± 1.522
Positive attitude (drug abuse)	157 (49.7)	56 (17.7)	29 (9.2)	31 (9.8)	43 (13.6)	2.19 ± 1.47
Low self-confidence	133 (42.1)	57 (18)	39 (12.3)	35 (11.1)	47 (14.9)	2.37 ± 1.49
To eliminate shyness	180 (57.0)	47 (14.9)	24 (7.6)	47 (14.9)	18 (5.7)	1.97 ± 1.32
Parents’ divorce	108 (34.3)	37 (11.7)	44 (14)	70 (22.2)	56 (17.8)	2.77 ± 1.54
Lack of amusement facilities	164 (51.9)	43 (13.6)	43 (13.6)	22 (7.0)	43 (13.6)	2.16 ± 1.46
Disability in resolving routine problems	139 (44.0)	59 (18.7)	49 (15.5)	14 (4.4)	55 (17.4)	2.32 ± 1.49
Crowded family	185 (58.5)	55 (17.4)	33 (10.4)	19 (6.0)	24 (7.6)	1.86 ± 1.26
Having strict parents	153 (48.6)	45 (14.3)	45 (14.3)	24 (7.6)	48 (15.2)	2.26 ± 1.49
Presence of an addicted person in the family?	131 (41.5)	34 (10.8)	50 (15.8)	51 (16.1)	50 (15.8)	2.54 ± 1.53
Friends offer	108 (34.2)	45 (14.2)	31 (9.8)	55 (17.4)	77 (24.4)	2.83 ± 1.62
Family disputes	133 (42.1)	37 (11.7)	36 (11.4)	48 (15.2)	62 (19.6)	2.85 ± 1.60
Access to drugs *	114 (36.2)	37 (11.7)	39 (12.4)	59 (18.7)	66 (21.0)	2.76 ± 1.59
Lack of access to physician consultation	145 (45.9)	49 (15.5)	37 (11.7)	39 (12.3)	46 (14.6)	2.34 ± 1.50
Low cost of drugs	149 (47.2)	64 (20.3)	44 (13.9)	28 (8.9)	31 (9.8)	2.13 ± 1.35
Having free time	146 (46.2)	54 (17.1)	42 (13.3)	34 (10.8)	40 (12.7)	2.26 ± 1.44
Presence of an addicted person in a residential/educational place	155 (49.1)	50 (15.8)	45 (14.2)	29 (9.2)	37 (11.7)	2.18 ± 1.42
Others	288 (91.1)	14 (4.4)	3 (0.9)	5 (1.6)	6 (1.9)	1.18 ± 0.70

Note: * missing response.

**Table 3 medicina-58-00322-t003:** Student perceptions about physical or psychological effects/symptoms felt after abusive drug or illicit substances.

Statements	Agree *n* (%)	No Comment *n* (%)	Disagree *n* (%)
Transient euphoria	240 (75.9)	61 (19.3)	15 (4.7)
Improved memory and learning ability	109 (34.5)	71 (22.5)	136 (43)
Depression	197 (62.3)	55 (17.4)	64 (20.3)
Improvement in some somatic diseases	145 (45.9)	101 (32.0)	70 (22.2)
Increased self-confidence	132 (41.8)	85 (26.9)	99 (31.3)
Better acceptability by friends	118 (37.3)	50 (15.8)	148 (46.8)

**Table 4 medicina-58-00322-t004:** Association between attitude score and student’s characteristics.

Participants Characteristics	Mean	Std. Deviation (Std)	*F* Value	*t*-Value	*p*-Value
Family size					
1–4	50.91	16.28	5.188		0.006
5–9	50.98	18.74			
>10	61.59	12.39			
Course of study					
Pharmacy	51.09	18.85		−1.120	0.264 *
Nursing	53.40	16.49			
Father education					
Illiterate	51.57	17.40	3.451		0.017
Able to read and write	63.46	8.84			
School education	54.34	16.70			
University education	49.96	18.72			
Father occupation					
Clerk	50.05	19.89	2.342		0.055
Self-employed	58.85	15.60			
Unemployed	53.25	18.66			
Retired	51.31	16.12			
Others	58.10	9.39			
Smoking					
Yes	51.63	18.95		−349	0.727 *
No	52.36	17.20			
Received any course about drug use in the college *					
Yes	52.64	18.85		0.996	0.320 *
No	50.28	14.52			

* student’s *t*-test.

## Data Availability

Data are contained within the article.

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
