# Peer review of "Attitudes and Associated Demographic Factors Contributing towards the Abuse of Illicit Drugs: A Cross-Sectional Study from Health Care Students in Saudi Arabia"

_medicina, 2022, doi:10.3390/medicina58020322_

Round 1

Reviewer 1 Report

Title: Attitudes and Associated Demographic Factors Contributing 2 Towards the Use of Abused Drugs: A Cross-Sectional Study 3 from Health Care Students ‘Perspectives´

Authors:   Iqbal et al.

The paper is on a cross-sectional survey in n = 184 pharmacy and n = 132 nursing studies regarding their attitudes toward use of illicit drugs.  Items with a rating included offer from friends, joy seeking and parent’s divorce and having access to drugs. Further, transient euphoria followed by depression are the most prevalent consequences of drug use. The authors conclude that some of the attitudes and consequences may be controllable or reversible. The identification of contributing factors may help direct government efforts towards improving access of public to mental health services and community care.

Also, to my knowledge, this is the first study on risk factor from Saudi Arabia on attitudes on and consequences of drug use. Interviews have been adapted to Arab language. The paper is in general well-written and structured, while some language editing would be helpful.

Some issues should be considered:

Major issues.

  1. It is certainly a weakness of the study having not recruited a comparison group from normal population in the same age group to compare attitudes and consequences. This would certainly help to understand whether the student population is special in their ratings or not. This and other potential shortcomings of the study should be mentioned in the paragraph on limitation of the study (already there) at end of the discussion section. 
  2. Regarding study design, it should be made very clear throughout the paper, that pharmacy and nursing students are statistically compared. Otherwise, the design of the study and the paper is unclear. Currently, descriptive statistics (attitudes) change with comparisons between groups (?). In this sense the paper could be more straightforward.

Minor issues:

  1. Abstract: did any of the participants meet criteria of harmful use or dependence? Please mention in the abstract and main text.
  2. Methods and material: it is not clear to the reader, on what study, OR or level of significance the sample size estimation is based on. Please explain.
  3. Data analysis: „The one-way ANOVA test was used, as appropriate, to assess the association between demographic characteristics“: were variables tested for deviation from normal distribution? If they deviated, non-parametric test statistics have to be used.
  4. Table 2: „psychiatric disorders“ -> mental disorders.
  5. Table 4: Please explain, what groups are compared (pharmacy vs. nursing students or Attitudes across sociodemographic categories)?  For some of the latter categories, student’s t-test (and not one-way ANOVA) is appropriate (e. g. smoking vs. non-smoking).

Author Response

Reviewer- 1

The paper is on a cross-sectional survey in n = 184 pharmacy and n = 132 nursing studies regarding their attitudes toward use of illicit drugs.  Items with a rating included offer from friends, joy seeking and parent’s divorce and having access to drugs. Further, transient euphoria followed by depression are the most prevalent consequences of drug use. The authors conclude that some of the attitudes and consequences may be controllable or reversible. The identification of contributing factors may help direct government efforts towards improving access of public to mental health services and community care. Also, to my knowledge, this is the first study on risk factor from Saudi Arabia on attitudes on and consequences of drug use. Interviews have been adapted to Arab language. The paper is in general well-written and structured, while some language editing would be helpful.

Response: Dear respected editor and team, thank you very much for your comment and we appreciated it  

Some issues should be considered:

Major issues.

It is certainly a weakness of the study having not recruited a comparison group from normal population in the same age group to compare attitudes and consequences. This would certainly help to understand whether the student population is special in their ratings or not. This and other potential shortcomings of the study should be mentioned in the paragraph on limitation of the study (already there) at end of the discussion section. 

Response: First and foremost, I thankful to you for the time to review. This study mainly focused to assess the perceptions of health care students, we have compared our findings between the health cares students (nursing and pharmacy). We agreed that we have not included normal population, since our objective is to include only students, since most of the students, getting lower performance in their courses and most of them were using prohibited substance (according to previous study) so we thought to restrict this study to students and we agreed that we will include one of the limitation as we have not included general population

Regarding study design, it should be made very clear throughout the paper, that pharmacy and nursing students are statistically compared. Otherwise, the design of the study and the paper is unclear. Currently, descriptive statistics (attitudes) change with comparisons between groups (?). In this sense the paper could be more straightforward.

 Response: My apologies for the confusion, I have corrected the objective to The purpose of this study is to compare the attitudes, views, and factors that influence drug addiction among pharmacy and nursing students at a Saudi Arabian university.

Minor issues:

Abstract: did any of the participants meet criteria of harmful use or dependence? Please mention in the abstract and main text.

Response: My apologies for the confusion, the main purpose of this study is to assess the health care student’s attitudes towards factors contributing use of abused drugs. According to study 41.7% of total students agreed that offer from friends is the most important/ very important factor for using abusive, illegal or prohibited substances followed by joy seeking 129(40.8%), parents’ divorce 126 (39.8%), having access to drugs/ substances 125(39.5%), problems in family 110(34.8%), presence of an addicted person in the family 66(20.8%) and teenager’s curiosity 101(31.9%). We have not assessed the prevalence of abused drug among students, but we assessed their attitudes, towards factors contributing to use of abused drugs and their opinions about psychological changes after drug use.

Methods and material: it is not clear to the reader, on what study, OR or level of significance the sample size estimation is based on. Please explain.

Response: According to university data at the time of study there were approximately 500 students perusing their graduation in both colleges of nursing and pharmacy Raosoft sample size calculator (http://www.raosoft.com/samplesize.html) with a 95% confidence level and a pre-determined margin of error of ±5%. We assumed that the response distribution for each question would be 50% because we were unsure of the potential results for each question. The calculated sample size was 218, but we decided to survey at least 400 students in an attempt to ensure a higher reliability.

(sample size was calculated similar to previous studies which have used same criteria

  1. Samreen, S., Siddiqui, N. A., Wajid, S., Mothana, R. A., & Almarfadi, O. M. (2020). Prevalence and Use of Dietary Supplements Among Pharmacy Students in Saudi Arabia. Risk management and healthcare policy13, 1523–1531. https://doi.org/10.2147/RMHP.S256656)

Data analysis: „The one-way ANOVA test was used, as appropriate, to assess the association between demographic characteristics “: were variables tested for deviation from normal distribution? If they deviated, non-parametric test statistics have to be used.

Response: My sincere apologies for the confusion, the data was normal and it was normally distributed between the variables, that is the reason we have chosen parametric test (ANOVA)

Table 2: „psychiatric disorders “-> mental disorders.

Response: My sincere apologies for the confusion, I have corrected it.

Table 4: Please explain, what groups are compared (pharmacy vs. nursing students or Attitudes across sociodemographic categories)?  For some of the latter categories, student’s t-test (and not one-way ANOVA) is appropriate (e. g. smoking vs. non-smoking).

Response: Table -4 shows the total attitude score and association between demographic categories also student specialty (nursing and pharmacy). Also I agreed with you that we used t-test, to compare two groups, more than two, we used ANOVA. (indicated with *mark below the table)

Reviewer 2 Report

The Authors introduced a paper describing the attitudes of health care students towards the factors associated with the use of illicit or prohibited drugs and substances. Although this manuscript seems to be interesting, there are several issues that need to be answered before the paper is published.

  1. In the introduction section, the Authors provide statistics from 2017. However, within 3 years these could  have  changed dramatically. Therefore, is it possible for Authors to state some current data?
  2. Please indicate the period when the study was carried on.
  3. The Authors should provide both the excluding and including criteria allowing students to participate in the study.
  4. Figure 1: please provide the number of students withdrawn from the study as this information is missing.
  5. There is no information regarding the total number of students that use once/occasionally/chronically drugs of abuse. Please state this in the manuscript. In line with this, there is also no data about the type of the drug, while in the results the Authors provide some information on hashish. I'm not sure whether the answers are only hypothetical since there is written that students reported " improvement in some somatic diseases".
  6. There are some gramma and punctuation errors/mistakes
  7. Overall, the obtained data do not provide efficient information or even conclusion.

Author Response

Reviewer- 2  

The Authors introduced a paper describing the attitudes of health care students towards the factors associated with the use of illicit or prohibited drugs and substances. Although this manuscript seems to be interesting, there are several issues that need to be answered before the paper is published.

  1. In the introduction section, the Authors provide statistics from 2017. However, within 3 years these could have changed dramatically. Therefore, is it possible for Authors to state some current data?

Response: my apologies, I have updated recent data which is highlighted in red color

  1. Please indicate the period when the study was carried on.

Response: my apologies, I have updated recent data which is highlighted in red color

  1. The Authors should provide both the excluding and including criteria allowing students to participate in the study.

Response: Entry-level students, aged ≥ 18 years who expressed a willingness to complete the survey and from the college of pharmacy and Nursing were included in the study. Students below the age of 18 and students from other disciplines, were excluded from the study.

  1. Figure 1: please provide the number of students withdrawn from the study as this information is missing.

Response: Withdrawal from the study after looking at questionnaire n = 22

  1. There is no information regarding the total number of students that use once/occasionally/chronically drugs of abuse. Please state this in the manuscript. In line with this, there is also no data about the type of the drug, while in the results the Authors provide some information on hashish. I'm not sure whether the answers are only hypothetical since there is written that students reported " improvement in some somatic diseases".
  1. Response: We have not assessed the prevalence of abused drug among students, but we assessed their attitudes, towards factors contributing to use of abused drugs and their opinions about psychological changes after drug use and association between attitudes and student’s characters. In the results Furthermore, 58.9 % of the 186 students did not agree that occasional use of prohibited drugs was acceptable (Figure-2). However, 146 (46.2%) of students agreed that drugs like hashish are not addictive. (Figure-2). This is just their opinions about drugs like hashish.

Also in this findings Approximately one-third of participants145(45.9%) reported improvement in some somatic diseases (pain, distress), this is their perception about physical or psychological effects/symptoms felt after abusive drug or illicit substances.

  1. There are some grammar and punctuation errors/mistakes

Response: My apologies we have corrected this with the help native English speaker

  1. Overall, the obtained data do not provide efficient information or even conclusion.

Response: My apologies we have corrected and revised the whole conclusion

Round 2

Reviewer 1 Report

Table 4: T- and F-values 

Author Response

My sincerely apologize , I have separated both T and F values . 

Reviewer 2 Report

After corrections made by the Authors, the paper is now acceptable for publishing.

Author Response

-